# Revealing the Immune Heterogeneity between Systemic Lupus Erythematosus and Rheumatoid Arthritis Based on Multi-Omics Data Analysis

**DOI:** 10.3390/ijms23095166

**Published:** 2022-05-05

**Authors:** Yuntian Zhang, Tzong-Yi Lee

**Affiliations:** Warshel Institute for Computational Biology, School of Life and Health Sciences, The Chinese University of Hong Kong, Shenzhen 518172, China; 221059046@link.cuhk.edu.cn

**Keywords:** systemic lupus erythematosus, rheumatoid arthritis, single-cell RNA sequencing technology, immune repertoire, cell–cell interaction

## Abstract

The pathogenesis of systemic lupus erythematosus (SLE) and rheumatoid arthritis (RA) are greatly influenced by different immune cells. Nowadays both T-cell receptor (TCR) and B-cell receptor (BCR) sequencing technology have emerged with the maturity of NGS technology. However, both SLE and RA peripheral blood TCR or BCR repertoire sequencing remains lacking because repertoire sequencing is an expensive assay and consumes valuable tissue samples. This study used computational methods TRUST4 to construct TCR repertoire and BCR repertoire from bulk RNA-seq data of both SLE and RA patients’ peripheral blood and analyzed the clonality and diversity of the immune repertoire between the two diseases. Although the functions of immune cells have been studied, the mechanism is still complicated. Differentially expressed genes in each immune cell type and cell–cell interactions between immune cell clusters have not been covered. In this work, we clustered eight immune cell subsets from original scRNA-seq data and disentangled the characteristic alterations of cell subset proportion under both SLE and RA conditions. The cell–cell communication analysis tool CellChat was also utilized to analyze the influence of MIF family and GALECTIN family cytokines, which were reported to regulate SLE and RA, respectively. Our findings correspond to previous findings that MIF increases in the serum of SLE patients. This work proved that the presence of LGALS9, PTPRC and CD44 in platelets could serve as a clinical indicator of rheumatoid arthritis. Our findings comprehensively illustrate dynamic alterations in immune cells during pathogenesis of SLE and RA. This work identified specific V genes and J genes in TCR and BCR that could be used to expand our understanding of SLE and RA. These findings provide a new insight inti the diagnosis and treatment of the two autoimmune diseases.

## 1. Introduction

Systemic lupus erythematosus (SLE), characterized by immune complex formation and T cell infiltration into tissues, is an autoimmune disease which can cause organ damage [1]. Rheumatoid arthritis (RA), characterized by chronic synovitis inflammatory and bone erosion, is also an organ-specific autoimmune disease [2]. Both autoimmune diseases could be life-threatening and do great harm to people’s health. Several studies have already been conducted to disentangle the pathogenesis of these two autoimmune diseases [3,4,5,6]. Marion and Postlethwaite illustrated that over-activation of B lymphocytes with excess production of autoantibodies contributed to the pathogenesis of SLE [3]. McGonagle et al. proposed that the immune complex formation and pro-inflammatory cytokine production, including TNF-α and IL-6, are associated with the production of autoantibodies against citrullinated proteins in the joints and other positions [4]. Additionally, researchers have found that the loss of peripheral immune tolerance that encompasses reduced T-reg cells highlights the pathogenesis of both SLE and RA [5,6]. The roles of infections in SLE and RA were reported previously. The loss of tolerance could be triggered by viral and bacterial infections in genetically predisposed people. Bo M et al. have already demonstrated that IRF5 is a potential target of autoimmune response triggered by EBV and MAP in rheumatoid arthritis patients in comparison with the general population [7]. They also demonstrated the differential antibody responses to BOLF1, MAP_4027 and IRF5 peptides could be used as diagnostic biomarkers among connective tissue diseases (rheumatoid arthritis, RA; systemic lupus erythematosus, SLE) [8]. It was found that the T-cell receptors in the peripheral blood mononuclear cells of patients can be used as a hallmark to distinguish systemic lupus erythematosus and rheumatoid arthritis samples from healthy control samples [9]. Additionally, the B-cell receptor has also achieved recognition in the functionality of autoimmune diseases and B-cell-targeted therapies were reported to be successful in the treatment of SLE and RA [10,11]. However, the comparisons between SLE and RA in terms of TCR and BCR repertoire remain lacking. Nowadays, the development of high-throughput sequencing technology has provided us a tool to characterize the immune features of SLE and RA group peripheral blood and determine the heterogeneity of the two autoimmune diseases [12]. In this work, a computational method called TRUST4, which was designed to extract immune features from bulk RNA-seq data, was utilized to build both TCR and BCR repertoires for SLE, RA and HC samples [13]. This work described the characteristics of the TCR and BCR repertoires from the aspects of clonotypes, V and J gene usage and CDR3 amino acid sequences’ length in patients with SLE and RA.

Multiple types of immune cells play roles in the mechanism of the pathogenesis of SLE and RA and have already been described previously, including for T cells, B cells, monocytes, neutrophils, natural killer cells (NK), dendritic cells (DC) and multipotent mesenchymal stromal cells. Both abnormal levels of IFNs and alterations in IFN signaling pathways have been observed in rheumatoid arthritis (RA) and systemic lupus erythematosus (SLE), and aberrant IFN expression in several cell types was also reported [14,15]. Several alterations in immune cells characteristics were reported previously. The absolute number and frequency of NK cells were observed to be diminished in SLE patients [16]. CD4+ T cells are relevant to the apoptosis of SLE and responsible for the initiation and perturbation of RA pathogenesis [17]. The apoptosis of neutrophils in PBMC under both RA and SLE conditions is dysregulated, and the neutrophil could also be used as a potential target due to its specific features [18,19,20]. The contents of natural killer cells under RA conditions were observed to increase, and the interactions between NK and DC are thought to contribute to the pathogenesis of SLE [21]. Several cytokines and related signaling pathways have been reported to influence the pathogenesis of autoimmune diseases. Bo M et al. demonstrated that upregulated expression, the presence of microbial and endogenous ligands and increased sensitivity to TLRs signaling may confer a crucial role to TLRs in RA pathogenesis. It was found that TLRs are increased in both peripheral blood monocytes (TLR-2 and 4), synovial fibroblasts (TLR-3 and 7) and in synovial fluid macrophages (TLR-2 and 4) [22]. Moreover, tumor necrosis factor alpha (TNF-α) was reported to play a central role in the pathogenesis of RA by increasing inflammatory cytokine levels, activating macrophages and lymphocytes. They also proposed that bacterial and viral infections could increase the immune response via TLR-2 and TLR-4 binding, resulting in the TNF-α secretion by neutrophils. IL-2, as a pleiotropic cytokine indispensable to proper T reg cell function, was reported as a non-pancreatic autoimmune target in type 1 diabetes (T1D) [23]. Additionally, a reduction in peripheral IL-2 bioavailability could interfere with T-reg homeostasis and function and lead to broken immune tolerance. Thus loss of immune tolerance to IL-2 may lead to the dysfunction of T-reg cells and induce the pathogenesis of autoimmune diseases. Bo M et al. evaluated the levels of anti-IL-2 antibodies (Abs) against IL-2, viral epitopes and interferon regulatory 5 between RA patients and healthy controls [24]. They found Ab reactivity reached the highest levels for IRF5, EBV and IL-2 in RA with significantly lower values among HCs, which suggests possible cross-reaction between IRF5/EBV homologous antigens and shifts in T-cell balance disrupted by anti-IL-2 Abs. All these findings illustrate the specific alterations in distinct immune cells under SLE and RA conditions. Nevertheless, the interactions between different types of immune cells have not been well established. Macrophage migration inhibitory factor (MIF) was found to be a pro-inflammatory cytokine, and one recent study reported an increase in serum MIF levels in patients with SLE [25]. Galectin-9 (LGALS9) was reported to be a multifunctional immunomodulatory factor highly expressed in RA, and another recent study proved the positive correlation between Gal-9 expression and disease activity in RA patients [26]. However, both MIF and GALECTIN family gene expression and related ligand/receptor pairs in different types of immune cells under SLE and RA conditions have not been reported. Our study revealed changes in several immune cell proportions under SLE and RA conditions, compared to normal conditions, and identified the top 20 enrichment pathways in different types of immune cells under both SLE and RA conditions based on the integration of single-cell RNA-seq data. We used a cell–cell communication analysis tool called CellChat to detect changes of MIF and GALECTIN family gene transcriptional expression and relative ligand/receptor pairs [27].

To summarize, our study revealed the heterogeneity between SLE and RA based on both the construction of immune repertoire and single-cell RNA-seq data. Furthermore, our findings could be useful to the development of new biotherapy and contribute to our understanding of the mechanism behind the pathogenesis of SLE and RA. The flowchart for this paper is indicated in Figure 1.

## 2. Results

### 2.1. T-Cell Receptor Repertoire Analysis

The T-cell receptor repertoires of peripheral blood in the SLE group, RA group and HC group were extracted from bulk RNA-seq data via standard workflow of TRUST4 algorithm [13]. After getting the immune phenotypes of T-cell receptor, we compared the top 10 clonotypes between SLE, RA and HC groups. The frequency of the top 10 clonotypes in SLE (31.41%) does not show a significant increase over that of HC group (29.25%, *p*-value = 0.7783) and RA group (23.08%, *p*-value = 0.2775). At the same time, there was not a significant change between RA group and HC group in terms of the top 10 clonotypes’ frequency (*p*-value = 0.1625). A box plot of the top 10 high-frequency clonotypes for each sample is shown in Figure 2A. We also analyzed the distributions of CDR3 amino acid sequence length under each condition. The most common CDR3 amino acid sequence length for SLE, RA and HC were 15, 16 and 16, respectively. CDR3 amino acid sequence length between SLE group, RA group and HC group do not show a significant difference (Figure 2C). Both Shannon–Weiner entropy and InvSimpson index were used to assess the diversity of the CDR3 amino acid sequences. The diversity of CDR3 amino acid sequences in RA group shows a significant increase over that in HC group in terms of both Shannon entropy (*p*-value = 0.0016) and InvSimpson index (*p*-value = 0.0036, Figure 2B,D). However, there was not a significant change between SLE group and HC group in terms of Shannon entropy (*p*-value = 0.85) and InvSimpson index (*p*-value = 0.13).

In order to compare the frequency of TRBV genes and TRBJ genes between SLE group, RA group and HC group, we generated a usage frequency stacked figure based on the common usage frequencies of TRBV genes and TRBJ genes (Figure 2E,F). TRBJ2-6, TRBV10-2 and TRBV3-1 in SLE group showed a significant decrease compared with those in HC group, while TRBV11-2, TRBV27 and TRBV30 showed a higher frequency in SLE group than in HC group (*p*-value < 0.05, Appendix A). In RA group, TRBJ1-5, TRBV27, TRBV30 and TRBV5-5 were observed to show a higher frequency than those in HC group (*p*-value < 0.05) and no TRBV gene or TRBJ gene showed a lower frequency than that in HC group (*p*-value < 0.05). Apart from that, TRBJ1-6, TRBV10-3, TRBV29-1 and TRBV3-1 showed a lower frequency in SLE group than in RA group, while TRBJ2-3 and TRBV4-3 were observed to show a higher frequency in SLE group than in RA group (*p*-value < 0.05).

### 2.2. B-Cell Receptor Repertoire Analysis

We constructed B-cell receptor repertoire from bulk RNA-seq data via TRUST4 algorithm. After obtaining the immune phenotypes of B-cell receptors through standard workflow of TRUST4 algorithm, we compared the top 10 clonotypes between SLE, RA and HC groups. The frequency of the top 10 clonotypes in SLE group (7.04%) showed a significant decrease compared with that in HC group (15.31%, *p*-value = 0.0034) and that in RA group (11.82%) and also showed a significant decrease compared with HC group (15.31%, *p*-value = 0.02744). The frequency of the top 10 clonotypes in SLE group were lower (7.04%) than in RA group (11.82%, *p*-value = 0.03917). A box plot of the top 10 high-frequency clonotypes for each sample is shown in Figure 3A. We also analyzed the distributions of CDR3 amino acid sequence length under each condition. The most common CDR3 amino acid sequence length of B-cell receptors for SLE, RA and HC groups were 15, 16 and 16, respectively, which displays a similar distribution to those of T-cell receptors (Figure 3C). Shannon–Weiner entropy and InvSimpson index were used to measure the diversity of CDR3 amino acid sequences under SLE, RA and HC conditions. Shannon entropy of CDR3 amino acid sequences in SLE group was higher than in HC group (*p* = 0.035). However, this finding was not statistically significant for the InvSimpson index (*p*-value = 0.062, Figure 3B,D).

In order to compare the frequency of IGHV genes and IGHJ genes in the SLE group, RA group and HC group, we generated a usage frequency stacked figure based on the common usage frequencies of the V gene and J gene (Figure 3E,F; Appendix A). The proportions of IGHJ6, IGHV2-5, IGHV3-53, IGHV3-33, IGHV7-4-1, IGHV3-66 and IGHV2-26 in SLE group were higher than those in HC group, while the proportion of IGHV3-41 showed a significant decrease in SLE group compared with HC group. The proportions of IGHJ5, IGHV1-NL1 and IGHV4-30-4 were higher in RA group than in HC group, while the proportions of IGHV1-18, IGHV3-9, IGHV1-69, IGHV5-51, IGHV1-8 and IGHV3-41 were lower in RA group than in HC group. The proportions of IGHJ2, IGHV1-18, IGHV3-9, IGHV2-5, IGHV3-74, IGHV5-51, IGHV3-53, IGHV3-33, IGHV1-46, IGHV7-4-1, IGHV1-8 and IGHV2-70 were higher in SLE group than in RA group, while the proportions of IGHV3-30, IGHV3-13 and IGHV4-30-4 were lower in SLE group than in RA group.

### 2.3. Single-Cell RNA-Seq Analysis

To comprehensively understand the discrepancies in PBMCs from SLE, RA and HC groups, we analyzed the single-cell RNA-seq data of PBMCs from three healthy people and three SLE patients and one RA patient (all datasets were obtained from the Gene Expression Ominibus database). After stringent raw data preprocessing and using canonical correction analysis (CCA) method to removing batch effects, 39,446 cells were collected for further analyses (Figure 4A). After normalization and principal component analysis (PCA) to reduce the dimension, we used t-distributed stochastic neighbor embedding (t-SNE) method to visualize all cells after all cells were grouped into 23 clusters first. And then we used the “FeaturePlot” function to visualize all the marker genes’ expression levels in individual clusters (Figure 4C). These clusters were assigned to known cell lineages through marker genes (Figure 4B). We computed the proportion of each cluster under each condition (Appendix A). We generated a cell proportion stacked figure (Figure 4D). The proportions of CD8+ T cells and FCGR3A+ monocytes were significantly increased in RA group compared to HC group (*p*-value = 0.048; *p*-value = 0.005) and the proportion of CD4+ T cells was significantly decreased in SLE group compared to HC group (*p*-value = 0.003). The proportions of CD8+ T cells and natural killer cells were higher in RA group than in SLE group (*p*-value = 0.018; *p*-value = 0.010).

### 2.4. Differentially Expressed Gene Identification and Functional and Pathway Enrichment Analysis of Distinct Immune Cells

The top-featured genes in each cluster were displayed in the heatmap (Figure 5A). We listed the top 3 differentially expressed genes for each immune cell type as follows, which can be used for future research: (1) CD4+ T cell: LTB, IL7R and MAL; (2) CD8+ T cell: CD8B, NELL2 and CCR7; (3) CD14+ monocyte: S100A9, LYZ and S100A8; (4) FCGR3A+ monocyte: FCGR3A, HES4 and CDKN1C; (5) NK: NKG7, GZMB and GNLY; (6) B: MS4A1, CD79A and HLA-DRA; (7) DC: FCER1A, HLA-DPA1 and HLA-DQB1; and (8) platelet: PPBP, CLEC1B and GNG11.

Then we screened the differentially expressed genes for each immune cell type between the SLE group and HC group as well as the RA group and HC group, with parameters |log2FC| > 0.25 and FDR < 0.05, respectively. When comparing SLE group and HC group, we identified 195 differentially expressed genes for platelets, 24 for dendritic cells, 13 for B cells, 367 for natural killer cells, 20 for FCGR3A+ monocytes, 86 for CD14+ monocytes, 14 for CD8+ T cells and 29 for CD4+ T cells (Appendix A). When comparing RA group and HC group, we identified 23 differentially expressed genes for CD4+ T cells, 28 for CD8+ T cells, 35 for CD14+ monocytes, 27 for FCGR3A+ monocytes, 70 for natural killer cells, 38 for B cells and 33 for dendritic cells (Appendix A). No differentially expressed gene were detected in platelets (FDR < 0.05). Metascape (Metascape. Available online: http://metascape.org/ (accessed on 3 April 2019)) was used to analyze multiple differentially expressed gene lists individually [28]. The top 20 significantly enriched terms are shown in Figure 5B,E. Among the 20 enriched terms between SLE group and HC group, four pathways were shared by eight immune cell types, including “positive regulation of cell death”, “response to bacterium”, “inflammatory response” and “neutrophil degranulation”. Among the 20 enriched terms between RA group and HC group, two pathways were shared by seven immune cell types, including “network map of SARS-CoV-2 pathway” and “response to virus” [29]. The Circos plot showed how genes from multiple gene lists overlap from eight immune cell subsets. Figure 5C,D represents the overlap of differentially expressed genes between the immune cell types when comparing SLE group and HC group. Figure 5F,G represents the overlap of differentially expressed genes between the immune cell types when comparing RA group and HC group.

### 2.5. MIF Family Signaling Pathways Expression between SLE Group and HC Group

Previous study has found that migration inhibitory factor (MIF) is pleiotropic cytokine which can be involved in regulating multiple signal transduction pathways [30]. MIF was reported to interact with its proposed receptor, CD74, located on the cell surface of the MHC class II invariant chain which subsequently forms a signaling complex with the accessory protein CD44 [31]. MIF was also reported to interact with the chemokine receptors CXCR2 and CXCR4 in complexes involving CD74 [32]. It was found that serum MIF levels were increased in patients with SLE [20]. However, the interactions including MIF–(CD74 + CXCR4) between different types of cells under SLE conditions have not been well studied. It is expected that these changes under SLE conditions may result in the alterations of complex intercellular communication networks among different types of immune cells. Recently, Jin et al. have reported the CellChat program could be used to predict how cells work together to transmit signaling and coordinate activities [27]. CellChat was used to compare the signaling strengths of MIF family ligand/receptor pairs among different cell populations in SLE and HC groups (Figure 6B). It is inferred that cells in SLE group generally displayed lower levels of connectivity and decreased numbers of interactions between various cell types. The chord figure of MIF–(CD74 + CXCR4) ligand/receptor pair shows an overall reduction of interaction numbers under SLE conditions compared to HC conditions (Figure 6A). Compared to HC group, the interactions between CD4+ T cells and natural killer cells, FCGR3A+ monocytes and CD14+ monocytes were lost. The interactions between natural killer cells, CD8+ T cells and two types of monocytes were also lost. We used the “netVisual_heatmap” function to visualize the details of the predicted information flow among the individual cellular components in the PBMC under SLE conditions and HC conditions (Figure 6C). Then we used the “plotGeneExpression” function to visualize the expression levels of MIF family cytokines, including MIF, CD74, CD44 and CXCR4 (Figure 6D). We noticed that the expression levels of MIF in each cell cluster were increased in SLE group compared to HC group. This finding was consistent with previous knowledge that serum MIF levels are increased under SLE conditions [20]. The expression levels of CD74 also showed an overall increase in all the cell types of SLE group compared to HC group. However, the expression levels of CXCR4 and CD44 show an overall decrease in each cell cluster of SLE conditions compared to HC conditions, and the decrease in CXCR4 and CD44 in each cell group led to the decrease in MIF–(CXCR4 + CD74) and MIF–(CD74 + CD44) ligand/receptor pairs.

### 2.6. GALECTIN-9-Related Signaling Pathway Expression between RA Group and HC Group 

GALECTIN-9 (LGALS9) is a multifunctional member of the galectin family expressed in various cell types and can take part in cell proliferation, differentiation, inflammation and immune cell formation. LGALS9 has been reported to be positively related with disease activities in RA patients [21]. GALECTIN-9 could be used as a new biomarker for evaluating RA activity and therapeutic effect. Nevertheless, the interactions including LGALS9 and CD45 between different types of cells under RA conditions compared to HC conditions have not been well described. We used CellChat to characterize the interactions between different cell populations in both RA group and HC group. The chord plot shows that the expression levels of LGALS9 and CD45 between different immune cells showed an obvious increase in RA group compared to HC group (Figure 7A). The circle plot summarized the maximum number of interactions among individual cell types in RA group and HC group (Figure 7B). The most obvious feature is that there are interactions between platelets and other cell populations, such as CD14+ monocytes, FCGR3A+ monocytes and dendritic cells. Additionally, GALECTIN signaling pathway was significantly enhanced in RA group compared to HC group. We used the “netVisual_heatmap” function to visualize the details of the predicted information flow among the individual cellular components in the PBMC under RA group and HC group (Figure 7C). Then we used the “plotGeneExpression” function to visualize the expression levels of GALECTIN family cytokines, including LGALS9, PTPRC, HAVCR2 and CD44 (Figure 7D). LGALS9, PTPRC and CD44 were all observed to be expressed in platelets under RA conditions compared to HC conditions. Additionally, LGALS9 under RA conditions was observed to be expressed in both B cells and dendritic cells compared to HC group. Interestingly, HAVCR2 was expressed in natural killer cells instead of CD14+ monocytes and FCGR3A+ monocytes in RA group compared to HC group. All these findings are consistent with previous findings and reveal the specific changes in GALECTIN-9-related signaling pathway expression during the pathogenesis of RA.

## 3. Discussion

Herein, our study has illustrated several important findings related to SLE and RA gene expression heterogeneity. At first, our study made full use of bulk RNA-seq data and computational algorithm TRUST4 to construct both TCR and BCR repertoires. The high variability of the CDR3 region and the amino acid sequence differences are determined by T- and B-cell receptors’ V(D)J rearrangement [33]. Previous study shows that characteristics of TCR and BCR repertoires can potentially assist in understanding adaptive immunity in autoimmune diseases [34,35]. Our methods were proved to be efficient in extracting immune profiles from bulk RNA-seq data without the process of additional T-cell receptor or B-cell receptor sequencing and thus saved the cost of building immune repertoires of the two autoimmune diseases using Multiplex PCR or Target enrichment [36]. Our results revealed the significant divergence in terms of CDR3 amino acid sequence diversity between the RA group and HC group. Although the Shannon entropy and InvSimpson index between the SLE group and HC group did not show significant changes, the SLE group’s Shannon entropy and InvSimpson index exceed the HC group’s Shannon entropy and InvSimpson index on average, which corresponds to our previous hypothesis that autoimmune diseases could lead to the increase in specific CDR3 amino acid sequences. Our results also identified significantly changed TRBV, TRBJ, IGHV and IGHJ genes, which were seldom reported in previous studies. These findings fill the limitations in the understanding of TCR and BCR repertoires’ features and suggest an immune response to the common autoantigens in SLE or RA patients, which could assist in the development of targeted biotherapy and the diagnosis of SLE and RA.

We took advantage of current popular single-cell sequencing technology to integrate multiple scRNA-seq datasets [37,38]. Compared to the HC group, the SLE group and RA group showed various changes in the composition of immune cell subsets. The proportions of CD8+ T cells and FCGR3A+ monocytes in the RA group were significantly higher than those in the HC group. Additionally, the proportion of CD4+ T cells in the SLE group were significantly lower than in the HC group. When comparing the SLE group to the RA group, we found the proportions of CD8+ T cells and NK cells in the SLE group were lower than those in the RA group, and our findings are consistent with previous study [39]. We speculated targeted CD8+ T-cell and FCGR3A+ monocyte therapy could relieve the pathogenesis of RA, and improving the levels of CD4+ T cells exogenously could save the status of SLE patients. We detected the top marker genes for each cell cluster and found CD4+ T cells and CD8+ T cells as well as FCGR3A+ monocytes and CD14+ monocytes shared similar marker genes, which are consistent with normal cognition. Functional and enrichment analysis for CD4+ T cells, CD8+ T cells, B cells, DC, NK cells, platelets, CD14+ monocytes and FCGR3A+ monocytes illustrated immune-related pathways. Shared pathways, including “positive regulation of cell death”, “response to bacterium”, “inflammatory response” and “neutrophil degranulation” for the SLE group exist in all immune cell types. Two pathways, “network map of SARS-CoV-2 pathway” and “response to virus”, are shared by all immune cell types in the RA group. This may indicate the synergistic influence among all the immune cells in the peripheral blood of RA and SLE patients.

Previous bioinformatics analysis was mainly focused on the expression profile of type I IFN under SLE and RA conditions, and they showed high serum IFN-I expression in individuals with SLE and RA [40,41]. Nevertheless, they did not cover the specific signaling pathways’ role in the pathogenesis of SLE and RA. This work focused on the interactions between different immune cell types and revealed changes in MIF and GALECTIN family gene expression and related ligand/receptor pairs. We made full use of popular computational biology algorithms to characterize the interactions between different cell types. Genetic studies have identified associations between MIF polymorphisms and autoimmune diseases [42,43]. Nevertheless, few studies have comprehensively investigated the role of MIF family genes in the susceptibility and severity of SLE. LGALS9 (Gal-9) was reported to play an important role in the immunoregulation of RA [44,45]. The functions of Gal-9 are complicated and few studies cover the potential role of Gal-9 in the pathogenic mechanism of RA. We detected the phenomenon of several cytokine expression level decreases, including in CXCR4 and CD44 in the SLE group, which has not been reported previously. Additionally, we revealed strengthened interactions of the LGALS9–CD45 ligand/receptor pair between platelets and monocytes under RA conditions. Further work involving more in-depth experiments is needed to verify the function of those cytokines and ligand/receptor pairs in the pathogenic mechanism of SLE and RA.

However, our study had some intrinsic limitations. First, the immune changes in the peripheral blood of SLE and RA patients were detected, whereas the antigens leading to the changes in T and B cells remain elusive and need further validation experiments. Second, although we verified the importance of two ligand/receptor pairs, LGALS9–CD45 and MIF–CD74 + CXCR4), in the pathogenic mechanism of RA and SLE, respectively, detailed mechanisms were not verified by in vitro or in vivo experiments. These contents should be further investigated.

## 4. Materials and Methods

### 4.1. Data Collection

Bulk RNA-seq data: We downloaded gene expression profiles (RNA-seq) of each condition’s patients’ peripheral blood mononuclear cells, including SLE samples (SRR13988788, SRR13988789, SRR13988790, SRR13988791, SRR13988792, SRR13988793), RA samples (SRR13989061, SRR13989062, SRR13989063, SRR13989064, SRR13989065, SRR13989066) and HC samples (SRR13988787, SRR13988795, SRR13988796, SRR13988798, SRR13989070, SRR13989072) from Gene Expression Omnibus [46,47].

Single cell RNA-seq data: We downloaded PBMC scRNA-seq data from Gene Expression Omnibus (GEO) database. PBMC scRNA sequencing data of one RA patient, three SLE patients and three healthy control samples were obtained from GSM4819747, GSM4954811, GSM4954812, GSM4217720, GSM4954813, GSM5335490 and GSM5335491, respectively [48,49,50,51].

### 4.2. T-Cell Receptor and B-Cell Receptor Repertoire Construction and Analysis

In order to construct T-cell receptor (TCR) and B-cell receptor (BCR) repertoires, we used TRUST4, which was reported to be efficient in reconstructing immune receptor repertoires for both T cells and B cells from bulk RNA-seq data [13]. After following standard pipelines of TRUST4, the output data, which covered traditional TCR and BCR repertoire information, were obtained and statistical analysis was performed as follows: (1) We computed the relative frequency of the 10 most abundant clonotypes in the TCR repertoire and BCR repertoire and used Student’s *t* test to compare the differences between them under SLE, RA and HC conditions, respectively. (2) TCR β-chain and BCR heavy-chain are composed of variable region and constant region [52]. Variable region could detect and bind to antigens specifically. The variable region of the β chain consists of three gene segments called variable (V), diversity (D) and junctional (J). The encoding genes of variable region result from the event of V(D)J rearrangement. Thus, V gene and J gene combination could reflect the diversity of clonotypes in both T-cell receptors and B-cell receptors. The proportions of all kinds of TRBV, TRBJ, IGHV and IGHJ genes under SLE, RA and HC conditions were calculated and Student’s t test was used to find the significantly changed TRBV, TRBJ, IGHV and IGHJ genes. (3) InvSimpson index and Shannon entropy of TCR β-chain and BCR heavy-chain complementary determining region 3 (CDR3) amino acid sequences from TCR and BCR repertoires under SLE, RA and HC conditions were computed to evaluate the diversity of TCR and BCR CDR3 amino acid sequences. (4) TCR β-chain and BCR heavy-chain CDR3 amino acid sequence length distribution conditions between SLE, RA and HC groups were analyzed.

### 4.3. Single-Cell RNA-Seq Data Preprocessing and Cell Class Identification

After downloading scRNA-seq data from GEO website, R package Seurat (version 4.0.0) was used to preprocess scRNA-seq data [53,54] At first, we set the threshold as following to exclude some cells: (1) The number of genes in each cell was greater than 200. (2) The mitochondrial gene expression ratio was less than 5% for all the PBMC scRNA-seq data. Seven GEO series scRNA-seq datasets were normalized using the “NormalizeData” function, and 2000 highly variable genes were identified using the “FindVariableFeatures” function, respectively. Then, the canonical correction analysis (CCA) method with the “FindIntegrationAnchors” and “IntegrateData” functions were used to remove batch effects after integrating seven GEO series scRNA-seq data. Finally, we obtained a total of 39,446 cells, combining SLE, RA and HC condition PBMC scRNA-seq data.

Subsequently, we performed data scaling and used PCA method to reduce dimension, and the top 50 principal components were selected to perform downstream analysis. We used t-distributed stochastic neighbor embedding (t-SNE) algorithm to visualize and explore the data. Cell clusters were identified by the function “FindNeighbors” using the K-nearest neighbors (KNN) algorithm and the function “FindClusters” with a resolution of 0.5. 

We utilized already known marker genes to annotate cell clusters and the correspondent relationship is shown as follows: (1) CD4+ T cell (IL7R), (2) CD8+ T cell (CD8A), (3) CD14+ monocyte (CD14), (4) FCGR3A+ monocyte (FCGR3A), (5) nature killer cell (NKG7), (6) dendritic cell (FCER1A), (7) platelet (PPBP) and (8) B cell (MS4A1) [55,56].

### 4.4. Differentially Expressed Gene Identification and Functional and Pathway Enrichment Analysis

Differentially expressed genes between each cell cluster were identified by using the “FindMarkers” function in Seurat and we set min.pct = 0.25, logfc.threshold = 1.0, only.pos = TRUE, and only genes with *p*-value < 0.05 were retained. We used the “Doheatmap” function to plot the heat map of the differentially expressed genes for each immune cell cluster. Additionally, we identified DEGs of different immune cell subtypes in two groups, SLE vs. HC and RA vs. HC, respectively. We set min.pct = 0.25, logfc.threshold = 0.25, only.pos = TRUE, and only genes with *p*-value < 0.05 were retained here. Then the DEGs of each cluster in SLE vs. HC group and RA vs. HC group were imported into Metascape (Metascape. Available online: http://metascape.org/ (accessed on 3 April 2019)) for the Kyoto Encyclopedia of Genes and Genomes (KEGG) and the gene ontology analysis of biological processes and reactome pathway analysis. The analysis was performed with false discovery rate (FDR) < 0.01 as the cut-off value [28,29].

### 4.5. Cell–Cell Interaction Analysis

R package CellChat was employed to identify and visualize cell–cell interactions of distinct immune cells under SLE, RA and HC conditions [27]. We followed the official workflow and loaded the normalized count from Seurat into CellChat and applied the standard preprocessing steps, including the functions “identifyOverExpressedGenes”, “identifyOverExpressedInteractions” and “projectData”, with a default parameter set. This work selected the Secreted Signaling pathways and used the precompiled human protein–protein-interactions as a priori network information. The core functions “computeCommunProb”, “computeCommunProbPathway” and “aggregateNet” were applied using standard parameters and fixed randomization seeds. Finally, the function “netVisual_individual” was used to visualize the MIF–CD74 + CD44) and GALS9–CD45 ligand/receptor pairs’ signaling strength. After that, we utilized the function “plotGeneExpression” to visualize MIF and GALECTIN family cytokine expression levels in each cell type from SLE, RA and HC groups, respectively.

### 4.6. Statistical Analysis

Student’s *t* test was performed to test for: (1) the relative frequency of the 10 most abundant clonotypes between SLE group, RA group and HC group; (2) Shannon entropy and InvSimpson index of CDR3 amino acid sequence length between SLE group, RA group and HC group; (3) differentially expressed V genes and J genes between SLE group, RA group and HC group; and (4) the proportions of distinct immune cell types between the three groups. All statistical data were analyzed using R scripts (version 4.0.0).

## 5. Conclusions

In conclusion, this work describes the immune features of both TCR and BCR repertoires under SLE and RA conditions. Abnormal changes in terms of immune cell composition and individual clusters’ transcriptional profiles were detected in the SLE group and RA group, which highlighted the immune infiltration influences in the peripheral immune environment under SLE and RA conditions. We also identified the role of two ligand/receptor pairs, MIF–(CD74 + CXCR4) and LGALS9–CD45, in the pathogenic mechanism of SLE and RA at the single-cell level. These findings will facilitate our understanding of the molecular and cellular basis of peripheral immune cells in SLE and RA. To summarize, this work could contribute to the development of diagnostic methods and biotherapy aimed at both SLE and RA patients in future.

## Figures and Tables

**Figure 1 ijms-23-05166-f001:**
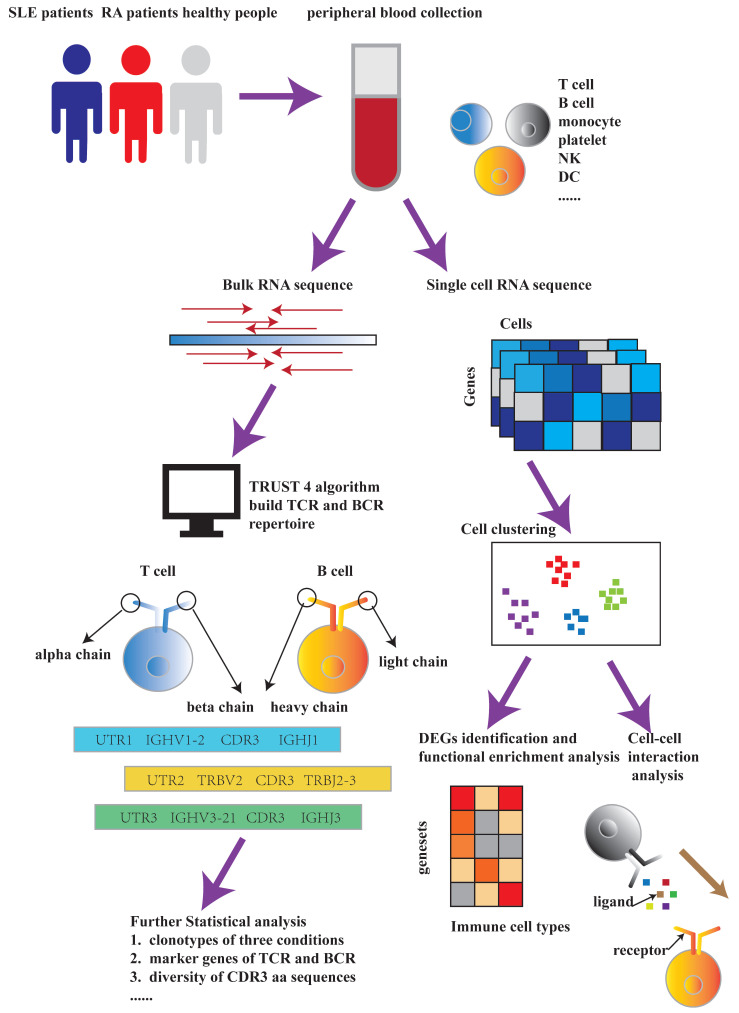
Flowchart for this paper. The main framework includes immune repertoire construction and single-cell RNA-seq data analysis. During the process of immune repertoire construction, bulk RNA-seq data were used as inputs for TRUST4 algorithm. After building both T-cell receptor and B-cell receptor repertoires, we used the extracted information to perform further statistical analysis. During the process of single cell RNA-seq data analysis, we utilized the original data to perform cell clustering and compared the proportions of various of immune cell types under SLE, RA and HC conditions. Finally, DEGs identification and functional enrichment analysis were conducted for different immune cell types, respectively, under SLE, RA and HC conditions. Cell–cell interaction analysis of MIF and GALECTIN family signaling pathways was also implemented between several immune cell types.

**Figure 2 ijms-23-05166-f002:**
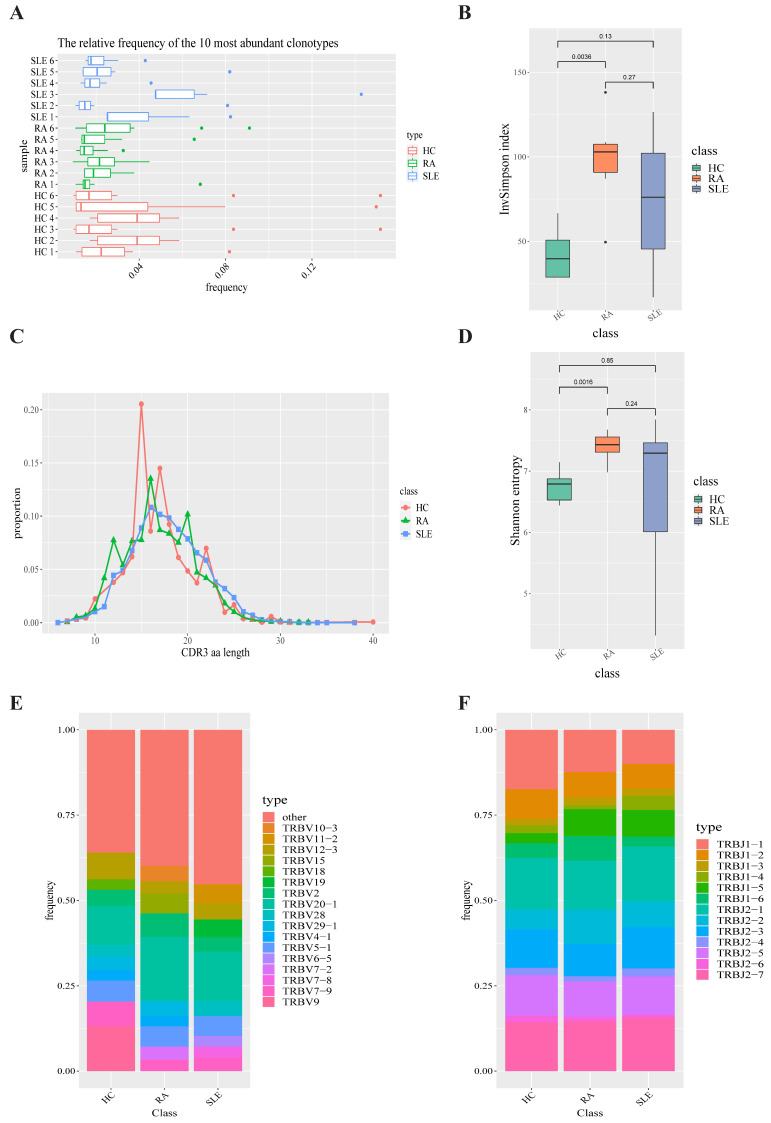
T-cell receptor repertoire construction and analysis. (**A**) Box plot showing the top 10 high-frequency T-cell clonotypes for each sample under SLE, RA and HC conditions in this study. (**B**) Box plots of Shannon–Weiner index for each sample under SLE, RA and HC conditions are used to compare the TCR diversity between SLE, RA and HC groups. (**C**) Distributions of TCR CDR3 amino acid sequence length in SLE, RA and HC groups. (**D**) Box plots of InvSimpson index for each sample under SLE, RA and HC conditions are used to compare the TCR diversity between SLE, RA and HC groups. (**E**) TRBV gene-usage-frequency stacked histogram showing the distributions of common TRBV gene in the SLE, RA and HC groups, respectively. (**F**) TRBJ gene-usage-frequency stacked histogram showing the distributions of common TRBJ genes in the SLE, RA and HC groups, respectively.

**Figure 3 ijms-23-05166-f003:**
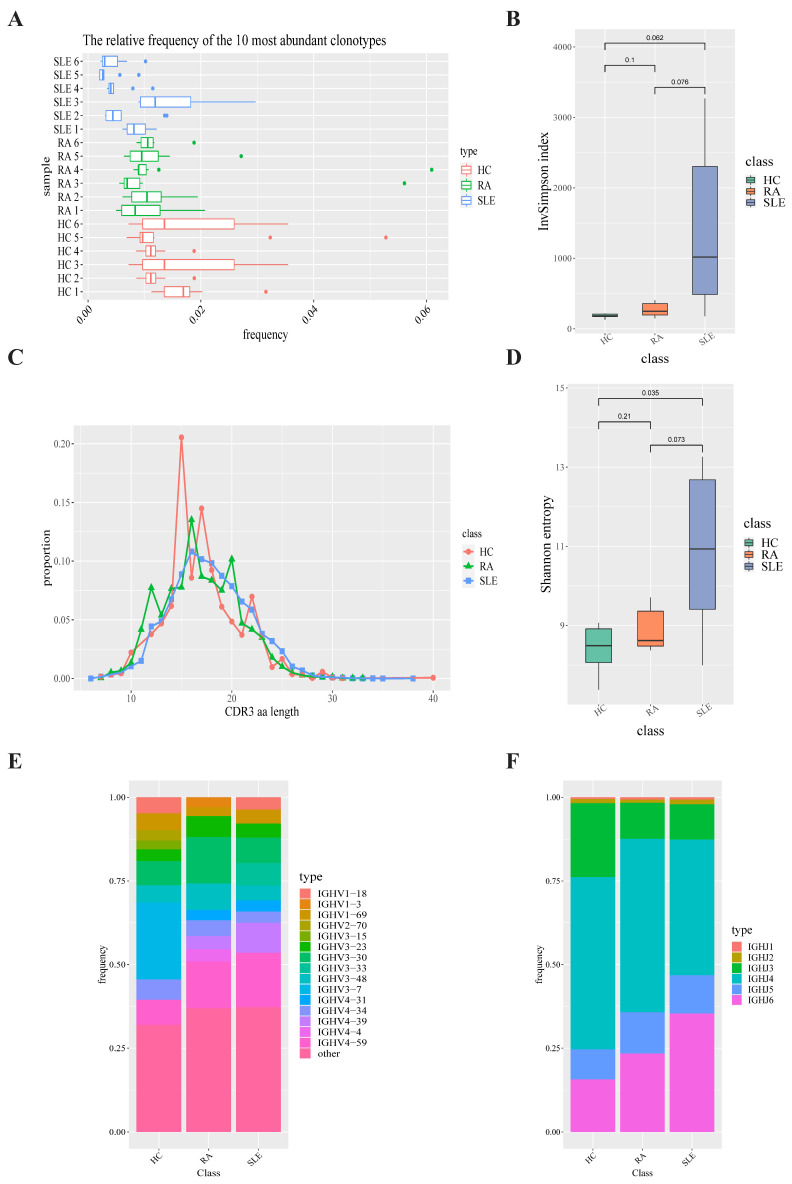
B-cell receptor repertoire construction and analysis. (**A**) Box plot showing the top 10 high-frequency B-cell clonotypes for each sample under SLE, RA and HC conditions in this study. (**B**) Box plots of Shannon–Weiner index for each sample under SLE, RA and HC conditions are used to compare the BCR diversity between SLE, RA and HC groups. (**C**) Distributions of BCR CDR3 amino acid sequence length in the SLE, RA and HC groups. (**D**) Box plots of InvSimpson index for each sample under SLE, RA and HC conditions are used to compare the BCR diversity between SLE, RA and HC groups. (**E**) IGHV gene-usage-frequency stacked histogram showing the distributions of common IGHV genes in the SLE, RA and HC groups, respectively. (**F**) IGHJ gene-usage-frequency stacked histogram showing the distributions of common IGHJ genes in the SLE, RA and HC groups, respectively.

**Figure 4 ijms-23-05166-f004:**
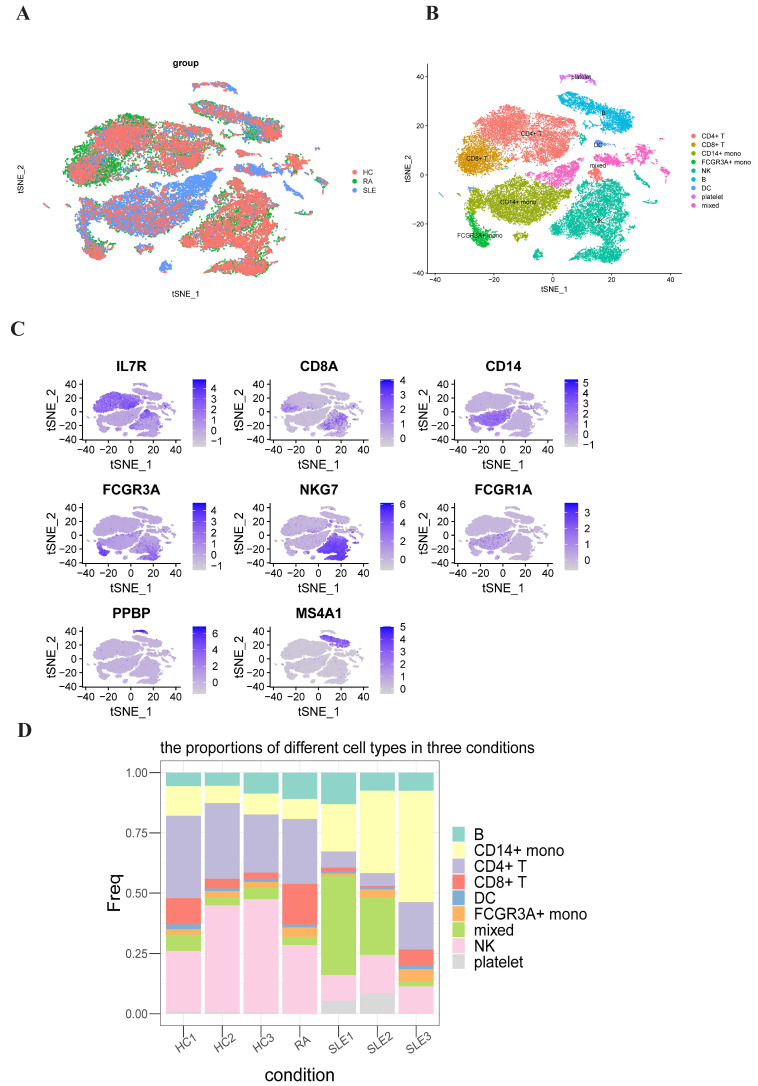
Single-cell RNA-seq data analysis of peripheral blood mononuclear cells (PBMC) in SLE, RA and HC groups. (**A**) The results of using canonical correction analysis (CCA) method to remove batch effects after integrating all the SLE, RA and HC scRNA-seq data. (**B**) TSNE plot representing the nine clusters across 39,446 PBMCs from seven samples (three SLE samples, three HC samples and one RA sample). (**C**) Representative marker genes define CD4+ T cells, CD8+ T cells, CD14+ monocytes, FCGR3A+ monocytes, natural killer cells, dendritic cells, platelets and B cells, respectively. (**D**) Bar plots showing the proportion of different immune cell types in each sample.

**Figure 5 ijms-23-05166-f005:**
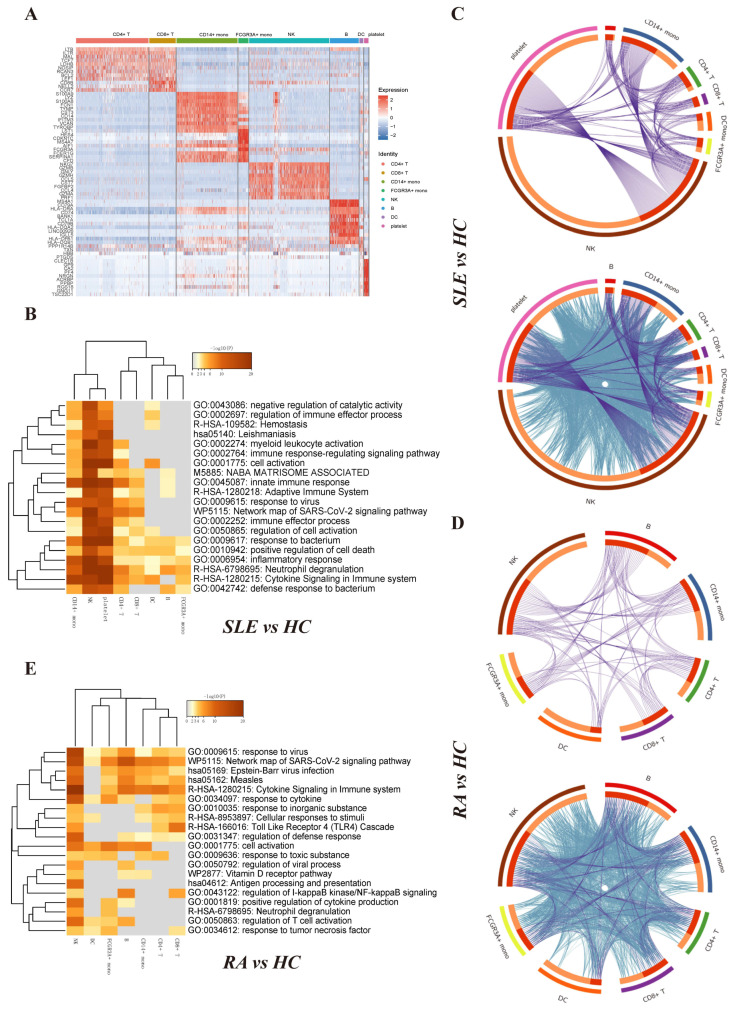
Differentially expressed gene identification and functional and pathway enrichment analysis of distinct immune cells. (**A**) Heat map of the top featured marker genes for distinct immune cell types. (FDR < 0.05, logFC > 1) (**B**) Heat map of the enriched functional and signaling pathways of differentially expressed genes between SLE group and HC group in distinct immune cell types. The heat map cells are colored according to the *p*-value of the enriched terms, and white cells indicate a lack of enrichment for that term. (**C**) The Circos plot shows how differentially expressed genes between SLE group and HC group from the given immune cell types overlap. Each arc represents each gene list’s identity. Purple lines link the same gene that is shared by multiple gene lists. Blue lines link the different genes where they fall into the same ontological term (the term needs to be statistically significantly enriched with a size no larger than 100). (**D**) The Circos plot shows how differentially expressed genes between RA group and HC group from the given immune cell types overlap. (**E**) Heat map of the enriched functional and signaling pathways of differentially expressed genes between RA group and HC group in distinct immune cell types.

**Figure 6 ijms-23-05166-f006:**
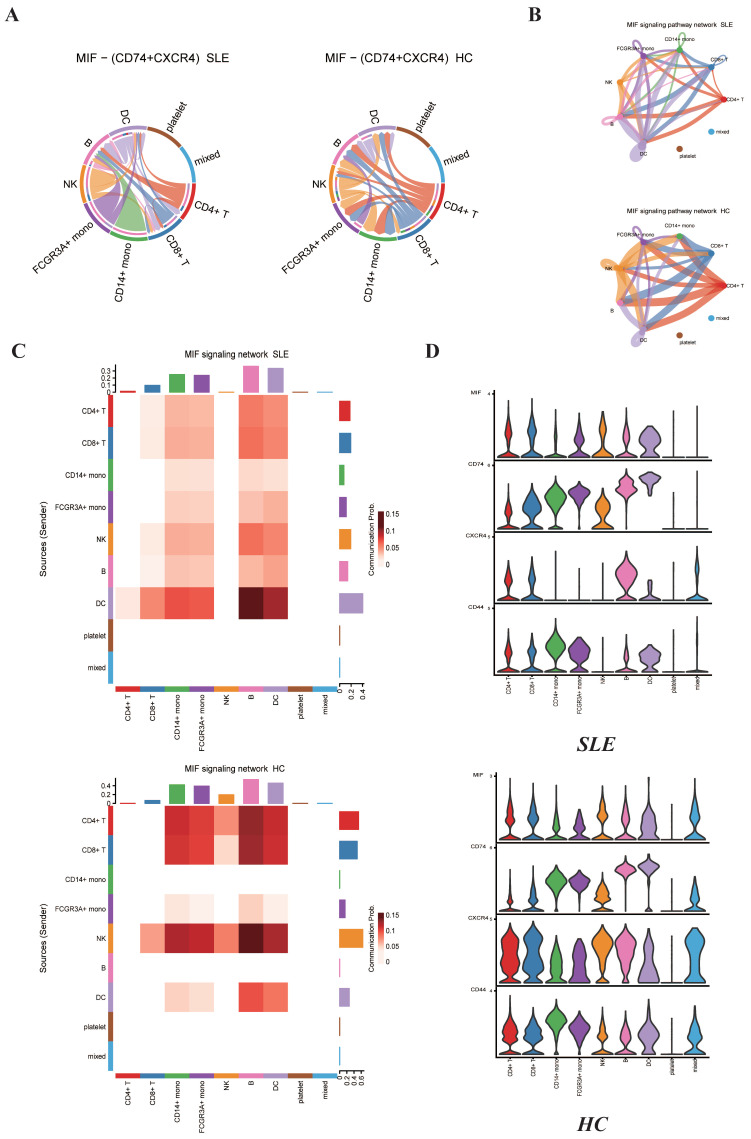
Cell-to-cell communications of MIF family signaling pathways among the distinct immune cells in the PBMC between SLE group and HC group predicted by the CellChat software. (**A**) Chord plots showing the interactions of ligand/receptor pairs MIF–(CD74 + CXCR4) between SLE group and HC group. (**B**) Circle plots summarizing the interactions of MIF signaling pathways among individual cell types in both SLE and HC groups. (**C**) Heat map showing the relative contribution of each cell type based on the computed four-network centrality measures of MIF signaling network between SLE and HC groups. (**D**) Violin plots showing the expression levels of MIF family cytokines in each immune cell type for both SLE and HC groups.

**Figure 7 ijms-23-05166-f007:**
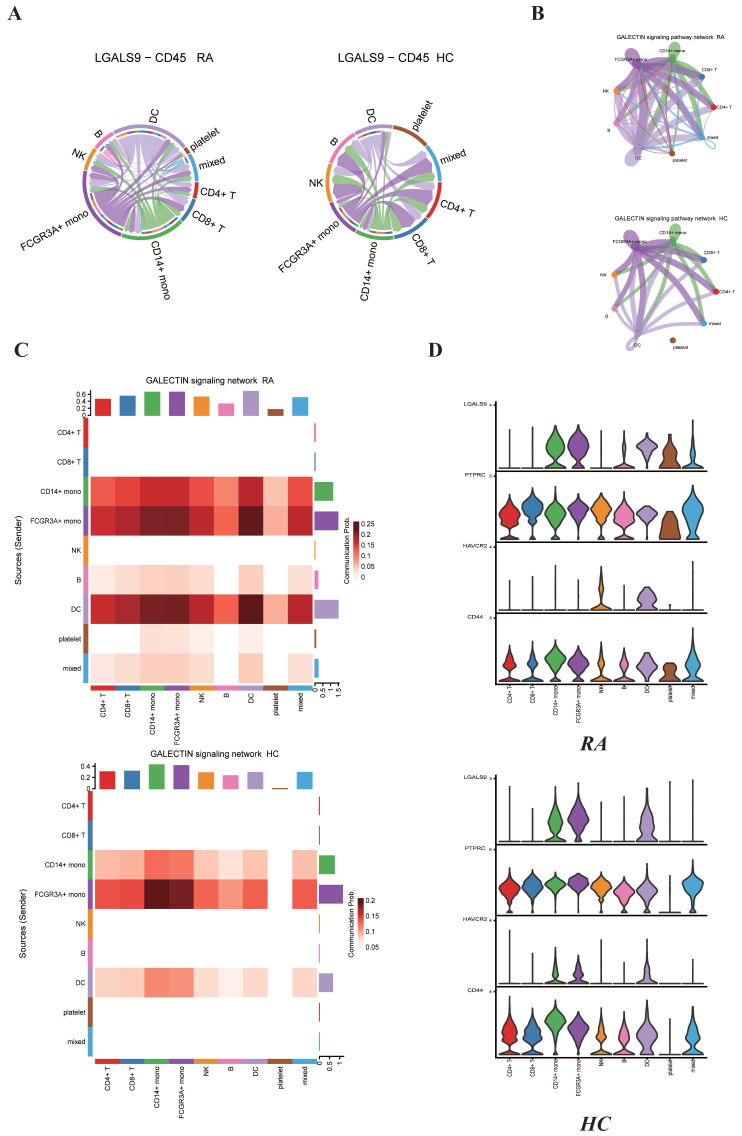
Cell-to-cell communications of GALECTIN family signaling pathways among the distinct immune cells in the PBMC between RA group and HC group predicted by the CellChat software. (**A**) Chord plots showing the interactions of ligand/receptor pair LGALS9–CD45 between RA group and HC group. (**B**) Circle plots summarizing the interactions of GALECTIN signaling pathway among individual cell types in both RA and HC group. (**C**) Heat map showing the relative contribution of each cell type based on the computed four-network centrality measures of GALECTIN signaling network between RA and HC groups. (**D**) Violin plots showing the expression levels of GALECTIN family cytokines in each immune cell type for both RA and HC groups.

## Data Availability

We analyzed publicly available datasets in this study. These data can be found here:
Transcriptome Analysis of PBMCs in peripheral blood of patients with hepatocellular carcinoma. Available online: https://www.ncbi.nlm.nih.gov/geo/query/acc.cgi?acc=GSE120663 (accessed on 7 March 2019)Transcriptome Analysis of PBMCs in peripheral blood of patients with RA. Available online: https://www.ncbi.nlm.nih.gov/geo/query/acc.cgi?acc=GSE169082 (accessed on 14 July 2021)Single cell RNA sequencing reveals cellular heterogeneity of PBMC of systemic lupus erythematosus patients. Available online: https://www.ncbi.nlm.nih.gov/geo/query/acc.cgi?acc=GSE162577 (accessed on 11 August 2021)Single cell RNA-seq for PBMC from rheumatoid arthritis patient. Available online: https://www.ncbi.nlm.nih.gov/geo/query/acc.cgi?acc=GSE159117 (accessed on 10 November 2020)Transcriptomic, epigenetic and functional analyses implicate neutrophil diversity in the pathogenesis of systemic lupus erythematosus. Available online: https://www.ncbi.nlm.nih.gov/geo/query/acc.cgi?acc=GSE139360 (accessed on 22 November 2019)Single-cell landscape of peripheral immune responses to fatal SFTS. Available online: https://www.ncbi.nlm.nih.gov/geo/query/acc.cgi?acc=GSE175499 (accessed on 14 August 2021) Transcriptome Analysis of PBMCs in peripheral blood of patients with hepatocellular carcinoma. Available online: https://www.ncbi.nlm.nih.gov/geo/query/acc.cgi?acc=GSE120663 (accessed on 7 March 2019) Transcriptome Analysis of PBMCs in peripheral blood of patients with RA. Available online: https://www.ncbi.nlm.nih.gov/geo/query/acc.cgi?acc=GSE169082 (accessed on 14 July 2021) Single cell RNA sequencing reveals cellular heterogeneity of PBMC of systemic lupus erythematosus patients. Available online: https://www.ncbi.nlm.nih.gov/geo/query/acc.cgi?acc=GSE162577 (accessed on 11 August 2021) Single cell RNA-seq for PBMC from rheumatoid arthritis patient. Available online: https://www.ncbi.nlm.nih.gov/geo/query/acc.cgi?acc=GSE159117 (accessed on 10 November 2020) Transcriptomic, epigenetic and functional analyses implicate neutrophil diversity in the pathogenesis of systemic lupus erythematosus. Available online: https://www.ncbi.nlm.nih.gov/geo/query/acc.cgi?acc=GSE139360 (accessed on 22 November 2019) Single-cell landscape of peripheral immune responses to fatal SFTS. Available online: https://www.ncbi.nlm.nih.gov/geo/query/acc.cgi?acc=GSE175499 (accessed on 14 August 2021)

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
