# Peer review of "Revealing the Immune Heterogeneity between Systemic Lupus Erythematosus and Rheumatoid Arthritis Based on Multi-Omics Data Analysis"

_ijms, 2022, doi:10.3390/ijms23095166_

Round 1
Reviewer 1 Report
The manuscript authored by Zhang et al. has aimed to reveal the immune heterogeneity between systemic lupus erythematosus (SLE) and rheumatoid arthritis (RA) using previously reported bulk and single cell RNA-seq data. The pipeline and data analysis methods used in this study were clearly explained. The study has detected dysregulated transcriptional profiles in various immune cell types of SLE and RA patients and functional and pathway enrichment analyses were made. Similar to the previous findings, MIF expression levels were found to be increased under SLE thus validating their findings. Likewise, increased GALECTIN-9 levels were found in immune cells associated with RA condition. This study has computationally identified the MIF-CD74+CXCR4 and LGALS9-CD45 interaction during SLE and RA pathogenesis respectively. It is good that the authors have understood the limitations of the study and have included their comments in their discussion. The conclusions were drawn appropriately from their findings and were not overstated.
Minor comments:
Typo error in line 96: "con- struction", there is no need for a hypen.
Line 127: The authors describe that the V and J genes reflect the diversity of clonotypes. It might be beneficial for the readers to state a brief background on the V and J genes and how it reflects the diversity of clonotypes. Although V(D)J rearrangement was stated in line 424, it may be helpful to describe why V and J genes are specifically analyzed.
Reviewer 2 Report
Dear authors,
please find the comments attached.
Kind Regards
